# The Role of Strength-Related Factors on Psychological Readiness for Return to Sport Following Anterior Cruciate Ligament (ACL) Reconstruction

**DOI:** 10.3390/healthcare11202787

**Published:** 2023-10-21

**Authors:** Oliver T. Lee, Mark A. Williams, Clare D. Shaw, Anne Delextrat

**Affiliations:** Department of Sport and Health Sciences and Social Work, Oxford Brookes University, Oxford OX3 0BP, UKwilliams.m@brookes.ac.uk (M.A.W.); clareshaw@brookes.ac.uk (C.D.S.)

**Keywords:** angle-specific torque, rate of force development, time since surgery, rehabilitation

## Abstract

Psychological readiness following anterior cruciate ligament reconstruction (ACLR) correlates with different return to sport outcomes. However, the relationship between strength and power and psychological readiness remains unexplored. The aim of this study was to investigate the relationship between anterior cruciate ligament return to sport after injury (ACL-RSI) scores and various hamstrings and quadriceps strength and power variables. Twelve participants (20.7 ± 2.5 years old; 174.2 ± 7.5 cm; 70.2 ± 8.5 kg; 18.2 ± 8.3% of body fat) who had an ACLR nine months or more before the study completed the ACL-RSI questionnaire and isokinetic strength testing of the hamstrings and quadriceps (60°·s^−1^ and 180°·s^−1^). Based on ACL-RSI scores, they were divided into “cases” and “controls”, deemed not psychologically ready and psychologically ready to return to previous sport performance (PILOS), respectively. The main findings are that quadriceps’ and hamstrings’ rate of torque development (RTD) and time since surgery were determinants of psychological readiness following ACLR. Furthermore, compared to controls, cases showed significantly lower quadriceps torque at angles close to full knee extension (40 deg and 30 deg from extension). They also showed lower RTD than controls, but no difference in peak torque. These results suggest that physiotherapists should facilitate athletes’ return to sport (RTS) by focusing on the restoration of RTD and strength at angles close to full knee extension.

## 1. Introduction

Anterior cruciate ligament injury (ACLI) is a common and incapacitating knee injury often sustained during sport participation and treated with ACL reconstruction (ACLR) followed by rehabilitation [1]. Return to sport (RTS) following ACLR is a major concern, with 73% of athletes able to participate in sport at some point following surgery, but only 49% returning to their pre-injury level of sport (PILOS) [2].

While the reasons why an athlete may not resume participation to their PILOS are multifactorial, the role played by psychological factors seems crucial [3]. Indeed, pooled data from 28 studies identified that out of 2918 participants, 65% cited a psychological reason for not returning to sport [4], the most significant factor consistently reported being that of fear of re-injury, with 24–50% of athletes reporting fear/ lack of confidence as their primary concern surrounding participation [5,6,7,8]. Physiological factors also play an important role in RTS, and in an attempt to understand the extent of the problem and synthesise current evidence for RTS decision-making, a group of expert clinicians produced a consensus statement in which a biopsychosocial model for RTS was developed [3]. This model emphasizes, in particular, the modifiable nature of an athlete’s psychological perceptions through relationships with the physiological determinants that may underpin an athlete’s confidence [3].

Sports with high occurrences of ACLI include team contact sports (i.e., basketball, football, hockey) and fixed-object high-impact rotational landing sports (i.e., gymnastics, obstacle course races) [9]. In addition to the characteristics of these sports that might lead to high-risk mechanisms for ACLI, athletes from these sports often show similar neuromuscular characteristics that could make them more at risk of ACLI [10,11]. The most commonly reported of these characteristics is a greater imbalance between the strength developed by the quadriceps and hamstrings in these athletes compared to athletes from other sports or non-athletic populations [10,11]. This imbalance is classically quantified by the functional hamstring-to-quadriceps ratio, defined as the ratio between hamstring eccentric peak torque (H_ecc_) and quadriceps concentric peak torque (Q_con_). Other characteristics of these athletes that might put them more at risk of ACLI include a low limb symmetry index (LSI) [10], in particular in sports with very specific roles for each leg (kicking in football, lunging in field hockey, for example), and the need for explosive force production and quick changes in direction [12].

Interestingly, the risk factors described above are also crucial factors to consider for RTS after ACLI. Indeed, deficits in the strength developed by the quadriceps and hamstrings are commonly reported in athletes, both at RTS and in the years following [13,14,15,16]. The hamstrings’ role as a synergist for the ACL is important in managing anterior tibial translation during landing, with maximal strength deficits reported as a risk factor for ACLI or re-injury in a number of prospective studies [17,18]. Quadriceps contractions help to absorb GRF during performance, in turn attenuating loads on lower extremities, such as the ACL during landing [19]. Deficits in quadriceps maximal strength have been associated with the incidence of re-injury following ACLI [20]. For both these muscle groups, it is commonly accepted that an LSI greater than 90% between operated and non-operated legs should be sufficient to reduce the risk of future damage to the surgical graft [20]. Despite this stipulation, many athletes still RTS with strength deficits in both muscle groups at RTS and in the years following, making appropriate rehabilitation essential [13,15,16].

More recently, authors have investigated quadriceps and hamstring force produced close to knee extension as risk factors because ACLI seem to occur at these angles [21,22], while peak values occur mid-range [23]. Hammond [24] identified that within a cohort of previously injured athletes who met initial criteria of LSI > 90% for maximal strength of the quadriceps and hamstrings, strength deficiencies were still observed in 72% of them, in particular at 10–40° from full extension. 

As mentioned earlier, the capacity to quickly produce force (rate of force or torque development, RFD or RTD) is a strong predictor of ACLI, in particular for the hamstring [24]. Indeed, most ACLI occurs within 0–61ms of initial contact with the ground [25]. The use of RFD, despite its relevance, remains very limited for RTS testing following ACLR [26], yet deficits in RFD are commonly seen following ACLR [27,28].

To our knowledge, only one study has examined the relationship between strength performance and psychological variables following ACLR utilising the anterior cruciate ligament return to sport after injury (ACL-RSI) questionnaire to quantify patient psychological readiness to RTS [29]. This study showed a significant association between ACL-RSI scores and hamstring peak torque, but not quadriceps, in a cohort of male athletes at 9 months post-surgery [29]. Additional findings from this study were the absence of significant differences in peak strength variables between a group with a low ACL-RSI score (<65) and a group with a high ACL-RSI score (>85). However, this study only measured peak strength, and there are currently no investigations on the association between ACL-RSI scores and other strength parameters, such as RFD or angle-specific torque.

Therefore, the primary aim of the present study was to investigate the strength and power determinants of athletes’ psychological readiness to RTS. A secondary aim was to compare strength and power performance of a group deemed psychologically ready for a return to PILOS against those deemed not psychologically ready to return to PILOS. We hypothesised that the group deemed not psychologically ready for a return to PILOS will show significantly lower strength and power values compared to the group deemed psychologically ready to return to PILOS.

## 2. Materials and Methods

### 2.1. Participants

Twelve participants, including six males and six females (20.7 ± 2.5 years old; 174.2 ± 7.5 cm; 70.2 ± 8.5 kg; 18.2 ± 8.3% body fat) volunteered to take part in the study. They were recruited from various university teams competing at regional to national levels. Inclusion criteria were having a primary ACLR more than 9 months prior to the start of the study because most athletes are cleared for RTS support between 6–9-month post-surgery. To be included in the study participants should also have already returned to some form of sport participation or have been officially cleared to do so by a qualified physiotherapist. Exclusion criteria were any current lower limb injury, any history of contralateral ACLI or recurrent ipsilateral ACLR, and any mental health condition, such as depression or anxiety, that could influence ACLI-RSI scores [30]. Participants gave written informed consent and the study was approved by the local university ethics committee (reference number 191305), in accordance with the principles set in the Helsinki declaration.

### 2.2. Procedures

The study design was cross-sectional, with participants attending the laboratory on one occasion only to perform psychological assessments first, followed by strength testing. The same investigator performed all the tests on all participants.

Psychological assessment consisted of the ACL-RSI questionnaire. This questionnaire was chosen for its very good validity and reliability [31,32]. In addition, its use was recommended in the 2015 consensus statement following ACLI [3]. The total score out of 100 was recorded, and it was also used to separate participants into one group deemed not psychologically ready to return to PILOS (cases) and a group deemed psychologically ready to return to PILOS (controls). The threshold used for these groups was a total score of 81.4%, as a previous study reported an association between scores <81.4% and the inability for athletes to return to this level [31]. 

Strength testing was undertaken on an isokinetic dynamometer (Biodex system 4; Biodex, Shirley, NY, USA). A standardised warm up was implemented prior to the initiation of isokinetic testing. This consisted of 8 min on a Monark cycle ergometer (874E; Monark, Vanberg, Sweden) at an intensity of 100 W, with three intermittent 6 s maximal effort sprints at minutes 6, 7, and 8. This was followed by five body-weight squats and two submaximal and three maximal counter-movement jumps [29]. After the warm-up, participants were instructed to sit upright on the chair with their hips flexed at 90°. Straps were secured to immobilise movement at the thighs, hips, and trunk in order to limit the influence of extraneous joint and muscle involvement. Straps were secured at the proximal tibias, from which participants would initiate/resist movement. Alignment of the lateral femoral condyle was made with the axis of the dynamometer level arm, with the proximal lever arm secured at the lateral malleolus. Participants were tested on both operated and non-operated legs and at two angular velocities, with all of these conditions randomised. The range of motion was from 0° (full knee extension) to 90° of knee flexion, and participants were given verbal encouragement to perform the fastest and strongest contraction effort throughout the range. The test consisted of concentric contractions of the quadriceps and eccentric contraction of the hamstrings at 60°·s^−1^ (three repetitions) and 180°·s^−1^ (five repetitions). Five repetitions have been recommended in previous studies at 180°·s^−1^ because it takes more repetitions to reach peak torque and it also induces less fatigue than slower velocities [24]. These velocities are characterised by excellent test–retest reliability (intraclass correlation coefficients of 0.95–0.98 [33]. Each condition was preceded by a familiarisation set.

The following variables were calculated as the average of the two best contractions at 60°·s^−1^ and the three best contractions at 180°·s^−1^:–Concentric peak torque of the quadriceps relative to body weight (Q_con,_ N·m·kg^−1^).–Eccentric peak torque of the hamstrings relative to body weight (H_ecc_, N·m·kg^−1^).–Peak H_ecc_:Q_con_.–Angle-specific H_ecc_, Q_con_, and H_ecc_:Q_con_ at 10° (H_ecc10_, Q_con10_, H_ecc_:Q_con10_), 20° (H_ecc20_, Q_con20_, H_ecc_:Q_con20_), 30° (H_ecc30_, Q_con30_, H_ecc_:Q_con30_), and 40° (H_ecc40_, Q_con40_, H_ecc_:Q_con40_) from full knee extension. These were calculated as averages between 0–10°, 11–20°, 21–30°, and 31–40°, respectively [34].–Quadriceps and hamstrings RTD in the first 50 ms and the first 100 ms (RTD_50_ and RTD_100,_ N·m·s^−1^) for H_ecc_, Q_con_, and H_ecc_/Q_con_, calculated as the ratio between the change in torque and the corresponding change in time in the first 50 ms and 100 ms of contraction, respectively. The onset of contraction was defined as a torque value of 1% of the peak torque produced during the same contraction [35]. The time windows of 50 ms and 100 ms were chosen as the best compromise between reliability and ecological validity. Indeed, ACL injuries usually occur in the first 50 ms after initial ground contact [36]. However, Mentiplay et al. [37] showed greater reliability of RFD at 100 ms than 50 ms. The same time windows were previously used in similar research [35].–LSI for all parameters: = ([(operated-non-operated)/non-operated] × 100.

### 2.3. Statistical Analyses

All data were presented as mean ± standard deviation, with a 95% confidence interval (CI). Statistical analyses were conducted using IBM SPSS V.29. The normality of data was assessed using the Shapiro–Wilk test. A stepwise linear regression was conducted on the entire sample to identify if strength data from the operated leg could be determinants of the ACL-RSI score. Subsequently, participants were divided into a ready to return to PILOS group and not ready to return to PILOS group. A mixed model analysis of variance (ANOVA) with repeated measures was performed to assess differences between groups (ready vs. not ready to return to PILOS) and angular velocities (60°·s^−1^ vs. 180°·s^−1^) on all outcome variables. Effect sizes were calculated as partial eta squared (ƞ^2^_p_) for the ANOVA and interpreted as no effect (0–0.05), minimum effect (0.05–0.26), and strong effect (0.26–0.64), while Cohen’s d (*d*) represented the effect size for post-hoc tests, and were interpreted as small (>0.2), medium (>0.5), and large (>0.8), [38]. The *p*-value was set at *p* < 0.05. 

## 3. Results

We recruited participants from football (*n* = 4), volleyball (*n* = 2), netball (*n* = 2), lacrosse (*n* = 2), tennis (*n* = 1), and triathlon (*n* = 1). Seven participants were identified as cases with ACL-RSI scores <81.4% and five participants were identified as controls with ACL-RSI scores > 81.5. Participant Demographics are summarised in Table 1.

The linear regression identified that three variables, namely quadriceps RTD_100_ at 60°·s^−1^, time since surgery, and hamstrings RTD_50_ at 60°·s^−1^, were significant predictors of the ACL-RSI score (r^2^ = 0.915). The statistics associated with each factor were beta = 0.583, t = 5.077 (*p* < 0.001) for quadriceps RTD_100_ at 60°·s^−1^; beta = 0.620, t = 5.855 (*p* < 0.001) for time since surgery; and beta = 0.411, t = 3.498 (*p* = 0.008) for hamstrings RTD_50_ at 60°·s^−1^.

Figure 1 and Figure 2 show the peak torque and angle-specific data for Q_con_ and H_ecc_. Unfortunately, we did not get enough data in the 0–10° range due to a lack of flexibility of some of our participants, and hence, these were removed from the analysis. The ANOVA showed a significant effect of angular velocity on Q_con_ peak torque (F(df) = 98.529(1), *p* = 0.001, ƞ^2^_p_ = 0.908), Q_con40_ (F(df) = 23.167(1), *p* = 0.001, ƞ^2^_p_ = 0.698), H_ecc30_ (F(df) = 7.012(1), *p* = 0.024, ƞ^2^_p_ = 0.412), and H_ecc20_ (F(df) = 5.817(1), *p* = 0.039, ƞ^2^_p_ = 0.393). Post-hoc analyses revealed significantly greater values at 60°·s^−1^ compared to 180°·s^−1^ in Q_con_ peak torque (95% CI: 2.33 to 3.15 N·m·kg^−1^ and 1.66 to 2.19 N·m·kg^−1^, respectively, for these two velocities, *d* = 1.46, Figure 1), Q_con40_ (95% CI: 1.40 to 1.73 N·m·kg^−1^ and 1.20 to 1.47 N·m·kg^−1^, respectively, for these two velocities, *d* = 0.87, Figure 1), H_ecc30_ (95% CI: 1.35 to 1.78 N·m·kg^−1^ and 0.98 to 1.66 N·m·kg^−1^, respectively, for these two velocities, *d* = 0.25, Figure 2), and H_ecc20_ (95% CI: 1.06 to 1.65 N·m·kg^−1^ and 0.68 to 1.46 N·m·kg^−1^, respectively, for these two velocities, *d* = 0.30, Figure 2). A significant group effect was also shown on Q_con40_ (F(df) = 4.975(1), *p* = 0.048, ƞ^2^_p_ = 0.345) and Q_con30_ (F(df) = 4.445(1), *p* = 0.049, ƞ^2^_p_ = 0.410). Significantly lower values were observed in cases compared to controls in Q_con40_ (95% CI: 1.12 to 1.49 N·m·kg^−1^ and 1.57 to 1.81 N·m·kg^−1^, respectively, for these groups, *d* = 0.79, Figure 1) and Q_con30_ (95% CI: 0.711 to 1.01 N·m·kg^−1^ and 1.12 to 1.49 N·m·kg^−1^, respectively, for these groups, *d* = 0.61, Figure 1). We did not find any significant interaction between groups and velocities on any torque variables (*p* > 0.05).

The statistical analysis showed a significant effect of angular velocity on H_ecc_ RTD_50_ (F(df) = 20.107(1), *p* = 0.001, ƞ^2^_p_ = 0.668) and H_ecc_ RTD_100_ (F(df) = 20.169(1), *p* = 0.001, ƞ^2^_p_ = 0.669). Post-hoc analyses revealed significantly lower values at 60°·s^−1^ compared to 180°·s^−1^ on H_ecc_ RTD_50_ (95% CI: 97 to 220 N.m.s^−1^ and 204 to 360 N.m.s^−1^, respectively, for these two velocities, *d* = 1.36, Figure 3) and H_ecc_ RTD_100_ (95% CI: 100 to 192 N·m·s^−1^ and 178 to 304 N·m·s^−1^, respectively, for these two velocities, *d* = 1.05, Figure 3). A significant group effect was also shown on Q_con_ RTD_100_ (F(df) = 4.679(1), *p* = 0.048, ƞ^2^_p_ = 0.338), with significantly lower values observed in cases compared to controls (95% CI: 763 to 1003 N·m·s^−1^ and 979 to 1315 N·m·s^−1^, respectively, for these groups, *d* = 1.42, Figure 4). We did not find any significant interaction between groups and velocities on any RTD variables (*p* > 0.05).

Regarding asymmetry between limbs, the ANOVA showed a significant group effect on LSI Q_con_ RTD_100_ (F(df) = 4.589(1), *p* = 0.045, ƞ^2^_p_ = 0.243) and LSI H_ecc_ RTD_50_ (F(df) = 6.224(1), *p* = 0.032, ƞ^2^_p_ = 0.384). Significantly greater values were observed in cases compared to controls on LSI Q_con_ RTD_100_ (95% CI: 0.96 to 1.15 and 0.83 to 1.00, respectively, for these groups, *d* = 0.59) and significantly lower values were observed in cases compared to controls on LSI H_ecc_ RTD_100_ (95% CI: 0.20 to 1.29 and 0.95 to 2.72, respectively, for these groups, *d* = 1.45). We did not find any significant interaction between groups and velocities on any LSI variables (*p* > 0.05).

## 4. Discussion

This study is one of very few observing the relationship between strength and power parameters and psychological readiness following ACLR. We measured a range of parameters during concentric contractions of the quadriceps and eccentric contractions of the hamstrings, since these antagonist muscles are involved in many sport movements. In particular, the hamstrings act as an ACL agonist by eccentrically contracting to decelerate the anterior translation of the tibia during powerful quadriceps contractions. The main findings are that the RTD of the quadriceps and hamstrings, as well as time since surgery, were the main determinants of psychological readiness following ACLR. In addition, compared to controls, cases showed significantly lower angle-specific Q_con_ close to extension and lower RTD for Q_con_, but no difference in peak torque values. 

We recruited athletes mostly from team contact sports, despite advertising the study to all university sports. It is not very surprising due to the high occurrence of ACLI reported in these sports [9]. It is well established that athletes from these sports show similar neuromuscular characteristics that could make them more at risk of ACLI, such as low H_ecc_:Q_con_ or high LSI [10,11]. 

A novel finding identified within this study is the extent to which RTD has been identified as both a determinant for psychological readiness and as a statistically significant difference between groups. This is the first study to examine the relationships between muscle RTD scores and psychological readiness, and, therefore, a comparison of results cannot be made against previous literature. The prevalence of RFD as a power deficit following ACLR, however, is well reported for the injured limb in the short term [27,28] but also in the longer term [39]. Deficits in RFD following ACLR could be due to both neural and peripheral muscular factors [39,40] and seem to closely reflect athletes’ rehabilitation status [39]. Several training modalities can improve RFD; for example, ballistic training performed around RTS [41] has shown greater increases in RFD than traditional strength and endurance training [42]. Sprint training also increased RFD (+29% in four weeks) in a recent study, while strength training did not [43]. With a potential overreliance on isolated maximal muscle strength after ACLR within traditional RTS training [41], results from this study indicate a greater focus should be given to training methods including a speed/power element in the facilitation of a successful RTS. Examples of such exercises include jump squats, mountain climbers, plank jacks, and skater hops for quadriceps power gain, and hurdle jumps, box jumps, jumping lunges, and resisted sprinting for hamstrings power gain. Physiotherapists should also monitor RTD (or RFD) during rehabilitation and consider it as an important variable for RTS. To explain the impact of RTD on psychological outcomes shown in the present study, the functional role played by the hamstrings and quadriceps in providing stability at the knee needs to be considered [44,45]. The ability of muscles to generate force within an early critical window (0–100 ms) is essential for stabilising the leg, attenuating GRF [19], and aligning landing mechanics within the period of time in which ACLI occur [25]. With residual movement impairments during RTS, it may be theorised that such neuromuscular deficits will affect the quality of movement athletes display [41], which may be negatively perceived by athletes, subsequently altering their ACL-RSI scores. These assumptions rely on the biopsychosocial framework, where an athlete’s physiological factors have a mediating role in their psychological perceptions [3]. However, further experimental studies should be conducted to confirm these associations. 

The role of peak torque as a determinant of ACL-RSI scores was not identified in the present study, suggesting that maximal force production might not be as important as sport staff (coaches, physiotherapists, and strength and conditioning staff) or researchers thought. This is partly in line with the findings of the only study in this area reporting no significant relationship between peak Q_con_ at 60°·s^−1^ and ACL-RSI scores in athletes nine months post-surgery [29]. However, the same authors also showed a small but significant correlation (r = 0.14) between peak H_ecc_ at 60°·s^−1^ and ACL-RSI scores [29], which is in contrast with our results. The study of O’Connor et al. [29] also separated participants according to their ACL-RSI scores, similar to the present study, and found no significant between-group differences in peak Q_con_ or peak H_ecc_, in accordance with our results. The contrasting results on H_ecc_ between the present study and the study of O’Connor et al. [29] could be explained by the different criteria used to separate groups (low (<65) and high (>85) ACL-RSI scores for cases and controls in the study by O’Connor et al. [29] compared to <81.4% and >81.4% in the present study). Having some individuals from different groups but with very close ACL-RSI scores in the present study could have biassed the results towards a lack of between-group differences. Other differences between studies are the significantly larger sample size in O’Connor [29]’s study (*n* = 452) and their cut-off at nine months post-surgery, compared to the range of longitudinal time frames examined within this study. It can be hypothesised that athlete perceptions within O’Connor’s [29] study are more of a reflection of their rehabilitation process. In contrast, for our participants who have returned to sport and, thus, benefit from greater sporting exposures, ACL-RSI scores illustrate the development of psychological barriers for athletes in the pursuit of a return to PILOS, which is a novel element of the current study. Finally, our data analysis was based on a linear regression, in contrast to correlations in the study of O’Connor et al. [29], which may explain the lack of association between some parameters and the ACL-RSI scores.

A novel finding of the present study is the significantly lower Q_con40_ and Q_con30_ in cases compared to controls (differences of 21% and 23%, respectively, at these angles). The role of the quadriceps at these shallow flexion angles is significant in relation to the number of ACLI that occur within them [21]. Previous studies have shown that individuals with weaker quadriceps have altered sagittal plane kinematics during landing and subsequently incur higher GRF upon a stiff-legged landing [19]. Similar to our comments on RTD, this process may result in a perception of instability during sport participation that could be reflected in the ACL-RSI scores. During rehabilitation, restoration of joint ROM is targeted during the early phases [46], yet little consideration is noted regarding muscular strength, specifically functionally important ranges (near full knee extension) in which non-contact ACLI occur (Hammond [24]). Our findings on angle-specific torque, together with the absence of significant differences between groups in peak torque, highlight the importance for physiotherapists to address strength deficits across the full range of motion and consider sport-specific mechanics (i.e., movements such as landing or side cutting, where the knee might not be very flexed, supporting stipulations made by Hammond [24], who similarly reported strength deficits between specific angles, where >90% LSI was achieved during peak torque assessments). The importance of the capacity to produce high torque when the knee is close to full extension has also been highlighted in a study on fatigue in female footballers, showing that H_ecc_:Q_con_ was significantly reduced after a simulated match only between 0 and 10° from full knee extension [34]. This highlights the reduced capacity of the hamstring to stabilise the knee joint close to extension in a fatigued state, where ACLI occurs, and highlights the importance of addressing residual strength deficits at these angles during rehabilitation. 

The limb symmetry index has been cited as a risk factor for lower limb injuries in previous literature [47], although there are contrasting results in the literature [48]. Our findings did not identify it as a major determinant of psychological readiness to RTS, suggesting that it might not be sensitive enough to affect athletes’ perceptions during exercise. However, we found that LSI for RTD_100_ in the quadriceps at both testing velocities was greater in cases than controls. This highlights one more time that power rather than strength could be an important factor that could play a role in athletes’ decisions to RTS. 

The main limitations of the present study are the lack of control over our participants’ rehabilitation and subsequent sport experience. Indeed, we could not capture the exercise types, volume, and frequency of rehabilitation sessions or past/current sport participation due to the long time lapse between the start of rehabilitation and testing for some of our participants. In addition, we chose to focus on psychological parameters as our main outcome and explore the role of strength and power variables as potential determinants of this. However, an argument can be made that constructs such as fear and confidence may instead impact physiological parameters through the avoidance of high-intensity rehabilitation required in order to efficiently address them. Finally, our sample size was small, in particular in comparison with the study by O’Connor et al. [29], which could limit the interpretation of our findings; however, this is counteracted by the fact that we observed relatively large effect sizes. We also recruited athletes from specific sports, which might limit the application of our findings to other sports. The main strengths of our study are the wide and novel range of outcome variables considered (in particular, RTD and angle-specific torque data) and the controlled laboratory conditions in which data was collected.

## 5. Conclusions

In conclusion, our findings showed that one of the main determinants of psychological readiness to RTS following ACLR was the RTD of the quadriceps and hamstrings. In addition, compared to controls, cases showed significantly lower angle-specific quadriceps torque close to extension and lower quadriceps RTD, but no difference in peak torque values. The importance of muscular strength across all angles of function was also identified, with quadriceps strength of particular importance at angles of functional relevance between 20° and 40° from full extension. These findings highlight the importance for physiotherapists to address muscular deficits across their full range (by choosing some exercises with the knee joint positioned near full extension) and develop power and explosiveness in addition to strength in rehabilitation. Future studies with larger sample sizes are needed to further investigate the relationship between strength and power and psychological factors playing a role in RTS.

## Figures and Tables

**Figure 1 healthcare-11-02787-f001:**
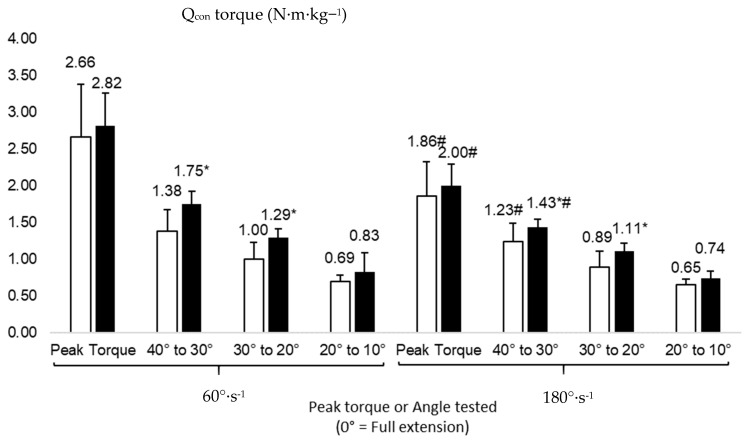
Concentric peak torque and angle-specific torque values of the quadriceps (Q_con_, N·m·kg^−1^) in cases (white) and controls (black). #: significantly different from 60°·s^−1^ at the same angle *p* < 0.05; *: significantly different from cases at the same angle (or peak torque), *p* < 0.05.

**Figure 2 healthcare-11-02787-f002:**
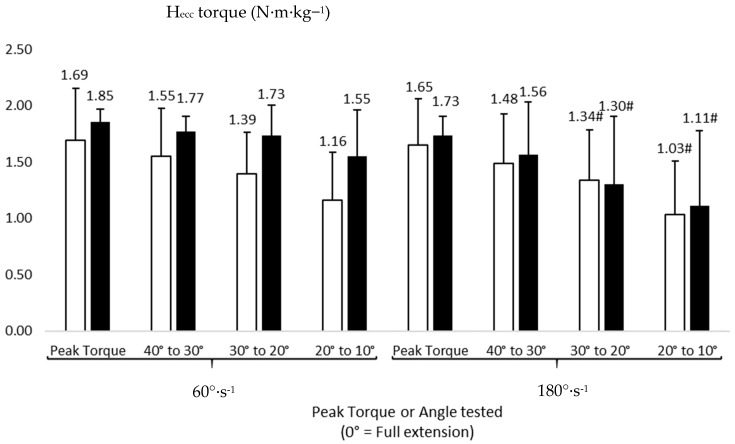
Eccentric peak torque and angle-specific torque values of the hamstrings (H_ecc_, N·m·kg^−1^) in cases (white) and controls (black). #: significantly different from 60°·s^−1^ at the same angle *p* < 0.05.

**Figure 3 healthcare-11-02787-f003:**
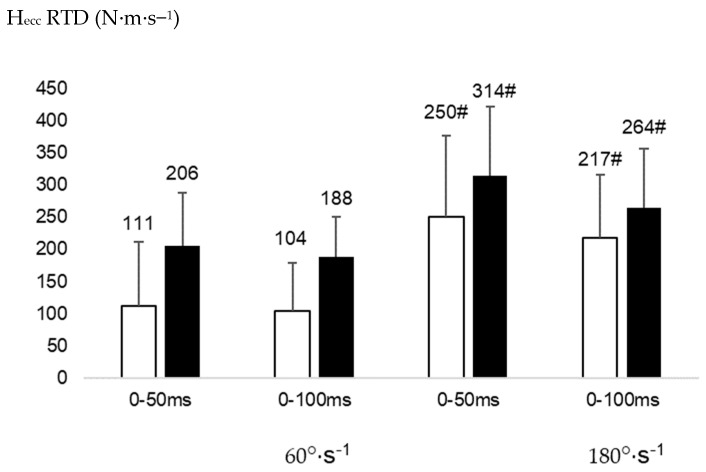
Rate of torque development (RTD) for the hamstrings eccentric (H_ecc_) contractions (N·m·s^−1^) between cases (white) and controls (black). #: significantly different from 60°·s^−1^ for the same time frame *p* < 0.05.

**Figure 4 healthcare-11-02787-f004:**
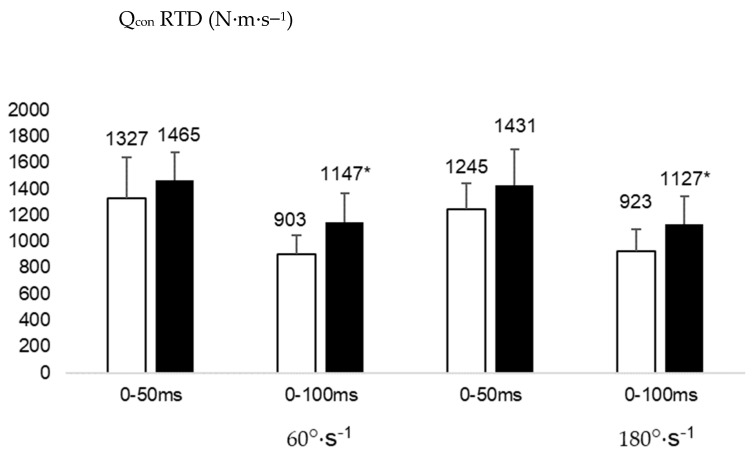
Rate of torque development (RTD) for the quadriceps concentric (Q_con_) contractions (N·m·s^−1^) between cases (white) and controls (black).*: significantly different from cases for the same time frame, *p* < 0.05.

**Table 1 healthcare-11-02787-t001:** Participants demographics (mean ± SD).

	All Participants (*n* = 12)	Cases ACL-RSI Scores <81.4% (*n* = 7)	Controls ACL-RSI Scores >81.4% (*n* = 5)
Age	20.7 ± 2.5	20 ± 2.2	21.6
Sex, male: female	6:6	3:4	3:2
Operated Side, right: left	7:5	4:3	3:2
Height (cm)	174 ± 8	170 ± 5	179 ± 8
Weight (kg)	70.2 ± 8.5	68.7 ± 7.5	72.26 ± 10.3
Body fat %	18.2 ± 8.3	20.9 ± 9.9	14.32 ± 3.5
Time Since Injury (months)	38 ± 21	33 ± 18	46 ± 24
Time Since Surgery (months)	31 ± 19	24 ± 13	41 ± 23
Time Since RTS (months) Reported to have returned to PILOS, yes:no	15 ± 15 5:8	11 ± 9 1:6	21 ± 21 4:1
Combined Injury, none: meniscal tear: Triad	3:6:3	3:3:1	0:3:2
Graft Type, HTG: PTG	12:0	7:0	5:0

Legend: Triad is reported with injury to each of the ACL, meniscus and associated medial collateral ligament or lateral collateral ligament. HTG: hamstring tendon graft, PTG: patella tendon graft, RSI: Return to Sport after Injury.

## Data Availability

Data from this study is available in Oxford Brookes University’s Institutional repository: accessed on 20 October 2023 https://radar.brookes.ac.uk/radar/items/fcc8c883-d87f-4ab3-bcce-047e56eef2df/1/.

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
