# Peer review of "The Role of Strength-Related Factors on Psychological Readiness for Return to Sport Following Anterior Cruciate Ligament (ACL) Reconstruction"

_healthcare, 2023, doi:10.3390/healthcare11202787_

Round 1

Reviewer 1 Report

Dear authors,

I am glad that I could read a good article and I can give you my opinion.

The quality of an article is also given by its ease of understanding, even if this article is specialized, which is why I recommend that you simplify, in expression, the discussions and conclusions. Thus, even less experienced readers can more easily understand the essence of your effort.

I think that the limitations of the study can be better and more widely described, and the strength of the study should be better highlighted.

I did not understand L92-94 "To be included in the study participants should also have already returned to some form of sport participation or have been officially cleared to do so by a qualified physiotherapist". this inclusion criterion is a bit vague. It is not specified what were the evaluation criteria targeted by the physiotherapist for patient inclusion and evaluation.

   The expression "some form of sport participation" is a bit vague!!!???

I hope my opinion and observations will help you.

Perhaps when you provide details to the other reviewers, you will reformulate the discussions and conclusions to be clearer, more precise and easier to understand for less specialized readers. But keep the central ideas that you have submitted to our attention.

Success

Reviewer 2 Report

Thank you very much for this work. The manuscript is sound and interesting. However, there are some points needed to be discussed.

Abstract

1.L. 18 the symbols for deg/s should be revised. 

2. L. 22 should be also revised to be a clearer statement 

Introduction

1. Please consider in L.57-58 this may be a grammatical error.

2. ACL-RSI in L. 72 was first cited in the text. Please provide the full word first here.

Methods

1. Was there the same investigators in this experiment?

2. L.127 please ensure whether or not CON testing was conducted for both Q and H muscles.

3. L.128 why at the faster test speed, needed to test for 5 repetitions.

4. Was there any specific command for RTD or power testing ?

5. Where abbreviation of some words is already cited, later it should consist to use abbreviations throughout the manuscript. Please kindly check.

Results

1. The title of Table 1 should be shorten and move to footnote of the Table

2. It would be better and clearer if Fig. 1&2 could present the angle-specific all along 0-90 deg of knee flexion. Now it also seems to lack of the data in the range of 0-10 deg as well. Both figures could also be improved by indicating tested angular speed perhaps in X axis. Please also move sig. Level into the Figure legends.

3. It may be clearer to insert Y axis titles for all figures.

Discussion

This part is well written and covered all points based on the study aim and findings.

1. Please kindly check possible mistake in L. 256 ( in week weeks)

2. It would be magnificent to sum up the rationale behind  for testing CON for quadriceps and ECC for hamstrings and clinical practicing to induce power gain in these patterns of muscle contraction.

Thank you.

Reviewer 3 Report

The main aim of the present study was to investigate the strength and power determinants, specifically those referred to hamstrings and quadriceps, that affect athlete’s psychological readiness to return to sport after suffering an anterior cruciate ligament reconstruction (ACLR). The findings revealed that quadriceps and hamstrings’ rate of torque development and time since surgery were the main determinants regarding psychological readiness after the injury. Moreover, those athletes deemed not psychologically ready to return to previous sport performance (i.e., cases) presented significantly lower angle-specific hamstrings and quadriceps torque close to extension and lower rate of torque development, without differences in peak torque, compared to those psychologically ready (i.e., controls). The authors concluded that restoration of rate of torque development and strength at angles close to full knee extension should be a priority for facilitating athlete’s return to sport.

The manuscript is well written and the topic interesting. Below, I provide some comments and suggestions for helping authors to improve the quality of their manuscript.

GENERAL COMMENTS:

Abstract:

Please try to limit abbreviations along the abstract section. Before using abbreviations, they should be previously described. For that purpose, previous description must be written in lower case except for proper names.

Line 25: RTS must be previously described.

Results:

Table 1: Please place Table legend at the foot of the Table.

Figures: For reporting significant differences between groups at specific angles/velocities, one asterisk is enough to mark these differences in one of the bars (controls or cases). Remove the others accordingly.

Figures 1 and 2: I assume that the bars on the left side of the Figures (from peak torque to 20 to 10°) refer to the velocity corresponding to 60°/s, while the right side of the Figure represents these values at 180°/s. Please include this information in both Figures for a better reader’s comprehension.

Line 225: Please change Qcon RTD to Hecc RTD

References:

Please revise this section carefully. There are few errors such as lack of publication year, name of the journal etc.

SPECIFIC COMMENTS:

Abstract:

Lines 22-23: In the Results section, Figure 1 shows no significant differences between cases and controls regarding angle-specific torque close to extension (see lines 176-178). Please clarify.

Introduction:

The introduction section is well written and provide scientific background to support the appropriateness of the present study. However, it would be interesting to describe those sport modalities that present a higher prevalence of ACLI. Similarly, I suggest authors to include some lines regarding sports-specific athletes’ characteristics that predispose them to suffer an ACLI. The authors are also encouraged to present the study’s hypotheses at the end of this section.

Materials and methods:

Participants

Why did you include athletes of these sport modalities and not from others (e.g., basketball, rugby, handball…). In this regard, what is the rationale for including athletes from a cyclic sport modality such as triathlon? Are these athletes really comparable? Scientific evidence suggest that acyclic sport modalities present a higher ACLI prevalence; similarly, following other taxonomies, different sport-specific characteristics  could influence ACLI prevalence (e.g., contact sports, limited contact sports, non-contact sports). It would be interesting to include some information regarding this issue in the Introduction and Discussion sections.

Statistical analyses

Since I assume that your sample followed a normal distribution, and therefore you conducted parametric tests, I think it would be interesting to run a repeated measures ANOVA with repeated measures in two factors: Group x Injury (i.e., operated vs non-operated) x Velocity (i.e., 60 vs 180). You could report more information from the pairwise comparisons and post-hoc test with this model.

Discussion:

You stated that one of the main findings of your investigation is that “…compared to controls, cases showed significantly lower angle-specific quadriceps and hamstrings torque close to extension and lower RTD…”. However, in the Results section (lines 176-178), you reported no significant differences for hamstring torque values at any angle or angular velocity (Figure 2). Please clarify and modify it accordingly.

Lines 333-335: Please revise these lines. See my previous comment and modify it accordingly.

Round 2

Reviewer 2 Report

Thank you very much for providing the revised MS. 

Author Response

Thank you for your comments

Reviewer 3 Report

I would like to congratulate the authors for the great job. The manuscript has substantially improved its quality. Well done!

I suggest only a couple of small modifications. Please, see below.

Abstract, lines 22-24: Since you did not find any significant interaction effect between groups and velocities, and you stated a significant group effect on both Qcon30 and Qcon 40, please remove “at 60°/s”.

Line 225: Please change “Figure 2” to “Figure 1”.

Author Response

Thanks, we have made the 2 modifications